# Predictors of loneliness among middle childhood and adolescence during the COVID-19 pandemic

Ashley Benhayoun[1], Anna Olsavsky[1], Terrah Foster Akard[2], Cynthia Gerhardt[1,3], Micah A. Skeens[1,3]*

**1** Center for Biobehavioral Health, Abigail Wexner Research Institute at Nationwide Children's Hospital, Columbus, OH, United States of America, **2** Vanderbilt School of Nursing, Vanderbilt University, South Nashville, TN, United States of America, **3** The Ohio State University College of Medicine Columbus, Columbus, OH, United States of America

* Micah.Skeens@nationwidechildrens.org

**Data Availability Statement:** Study data files and data dictionary are available on OSF (https://osf.io/8fec4/).

## Abstract

Social contexts (e.g., family, friends) are important in predicting and preventing loneliness in middle childhood (MC) and adolescence; however, these social contexts were disrupted during the COVID-19 pandemic. Comparison of social context factors that may differentially contribute to loneliness at each developmental stage (MC vs. adolescence) during the COVID-19 pandemic have been overlooked. This study examined longitudinal predictors of loneliness, including social contexts and COVID-19 impact, within MC (8-12y) and adolescence (13-17y). Parents reported on demographic information, and their children completed surveys on COVID-19 impact, loneliness, and family functioning using the COVID-19 Exposure and Family Impact Survey (CEFIS), the NIH Toolbox Loneliness (Ages 8-17) measure, and the PROMIS Family Relationships Short Form 4a measure, respectively. Regression models examined time one (T1; May-June 2020) predictors of time two (T2; November 2020-January 2021) MC child (*n*=92, M*age*=10.03) and adolescent (*n*=56, M*age*=14.66) loneliness. For the MC child model, significant predictors of higher loneliness included worse family functioning as well as higher COVID-19 impact and lower family income. On the other hand, higher adolescent loneliness was significantly predicted by not having married/partnered parents and was marginally significantly predicted by higher COVID-19 impact. The regression model with the full sample and interaction terms revealed no significant interactions, but that lower family functioning and higher COVID-19 impact were significant predictors of higher loneliness. Lower family income and lower in-person communication were marginally significant predictors of higher loneliness in the combined interaction model. Lastly, further exploratory mediation analyses displayed that family functioning significantly mediated the relationship between COVID-19 impact and T2 loneliness only for MC children and the full sample. Results support future interventions focused on optimizing family functioning to help mitigate MC loneliness in the context of adversity, such as a global pandemic.

**Funding:** Funding was provided through M.A.S' funding from the Nationwide Children's Hospital Intramural Funding Program. No funders were involved in study design or analysis, data collection, preparation of the manuscript, or decision to publish. https://www.nationwidechildrens.org/research/clinical-research/resources-for-investigators.

**Competing interests:** The authors have declared that no competing interests exist.

## Introduction

As a result of the COVID-19 pandemic, there have been surges in middle childhood (MC) and adolescent loneliness [1], potentially compounding known increases in loneliness during the transition from MC to adolescence [2, 3]. This is concerning, as loneliness during MC can potentially lead to negative outcomes such as suicidal behavior in adolescence [4] and conduct and internalizing issues in adulthood [5, 6]. Social contexts, including family environment and peers, have been shown to modify loneliness during the transition from MC to adolescence [7, 8]. Thus, it is important to understand predictors of loneliness during the challenging developmental transition from MC to adolescence, particularly in the context of the COVID-19 pandemic when MC children and adolescents had inhibited access to their social contexts. Further, the extent of loneliness disparities between the two developmental stages is not well-defined in the context of the COVID-19 pandemic. It is crucial to comprehend these distinctions as it may help shape interventions aimed at alleviating loneliness in both stages. Such interventions are currently limited, irrespective of the context, during or prior to the COVID-19 pandemic [9, 10]. Though studies have examined loneliness during the COVID-19 pandemic, they are limited by an absence of longitudinal projects that sample both MC children and adolescents [11], as well as a lack of studies that examine developmental differences in loneliness and the potential mitigating effects of social contexts. Thus, we examined the roles of family and peers on loneliness over time for the developmental stages of middle childhood (ages 8-12; referred to as MC/MC children) and adolescence (ages 13-17; referred to as adolescence/adolescents).

### MC loneliness and social contexts

MC loneliness has been linked to social contexts, and among these contexts, family holds significant influence [12, 13]. Low family cohesion and poor family functioning have been associated with MC loneliness, lower prosocial behavior, and overall worse mental health outcomes [14, 15]. Family hardships (e.g., family reorganization, economic disadvantage) are also difficult for MC children and are connected to elevated loneliness [16, 17]. Findings of these studies underscore the significance of favorable family relationships in mitigating MC loneliness and promoting overall well-being.

Social contexts outside the home are also crucial, as peer group acceptance progressively assumes greater significance during MC [18]. Peer rejection in MC is linked to increased loneliness [19]. Conversely, high quality friendships characterized by reciprocity are linked to improved psychosocial outcomes and serve as a protective factor, mitigating the adverse effects of solitude and relational aggression on depression [20, 21]. Although existing research suggests both family and peer social contexts independently impact loneliness in MC [14, 19], there is a lack of literature examining the concurrent influence of family and peer social contexts on MC loneliness, particularly during the COVID-19 pandemic.

### Adolescent loneliness and social contexts

Similar to MC, family context contributes to adolescent loneliness [22, 23]. High quality attachment and relationships with parents, along with positive family communication, are associated with lower adolescent loneliness and more favorable mental health outcomes [22, 23]. Adolescents with stable and secure attachment with their parents may be less likely to feel lonely during life transitions, such as moving to college, along with having healthy socioemotional adjustment and experiencing improvements in social competencies during these major life changes [24].

Outside of the household context, adolescents who feel connected to their peers and have higher quality friendships may experience lower depressive symptoms and lower feelings of

loneliness [8, 25]. In contrast, a lack of friendships, having undesirable peer relationships, and experiencing peer rejection and victimization are linked to higher loneliness [19, 26]. Having diverse sources of support, including parent, peer, and teacher support, are also associated with lower levels of loneliness [27]. These studies suggest that outside of the family unit, adolescents rely on their peers and various sources of support, mitigating their loneliness.

## Loneliness and the transition from MC to adolescence

The transition from MC to adolescence can be described as stressful, as it comprises many physical, emotional, and social changes [28]. Examples of these changes include working memory development [29], spending less time with family, and desiring more autonomy and individuation [30]. While navigating the transition from MC to adolescence, longitudinal studies have described the harmful trajectory of preexisting loneliness in MC persisting into adolescence and contributing to increased suicidal behavior and ideation [4, 31]. Likewise, longitudinal studies have reported increased internalizing problems and loneliness in the transition from MC to adolescence [2, 3]. Although the transition may be stressful, outside processes, such as poor family functioning and bullying, can exacerbate negative outcomes. For example, Hazel, Oppenheimer [32]) and Savell, Saini ([7]) report low parent–child relationship quality as a predictor of poor internalizing behavior outcomes in MC and adolescence. Similarly, MC children who are victimized by their peers face increased risk of internalizing issues in adolescence [33, 34]. The risk of poor wellbeing outcomes, including loneliness, during the transition from MC to adolescence is apparent, however limited data exists comparing the differences in the two developmental stages. It is known that contextual factors, such as family environment and peer relations, can ease or exacerbate the effects of the transition from MC to adolescence [35]. However, there is a dearth of literature examining these factors while comparing influences of loneliness between MC children and adolescents. Understanding differences and similarities in predictors of loneliness for the two developmental stages may be especially helpful to inform future intervention work to address the unique needs of both MC children and adolescents, as well as interventions aimed at the critical transition between these two developmental periods.

## MC and adolescent loneliness during the COVID-19 pandemic

The COVID-19 pandemic has disrupted the lives and affected the mental health of MC children and adolescents. Previous studies display that both MC children and adolescents have experienced lower health-related quality of life, higher rates of anxiety and depressive symptoms, and higher suicide risk [36, 37]. Increases in mental health symptoms in both developmental stages during the COVID-19 pandemic have been attributed to swift changes in daily life as a result of restrictions, such as increased social isolation, decreased physical activity, and increased exposure to pandemic-related information [38]. For example, in April-June 2020 when most school was closed or remote in the United States, adolescents reported that online schooling negatively impacted their ability to learn [39]. Both MC children and adolescents also reported feelings of anxiety, boredom, and increased parent stress and family conflict during initial lockdown and the following several months [39, 40]. Additionally, adolescents expressed life became more complicated, they lacked control, and they were missing social milestones [41].

Along with increased feelings of sadness, fear, and isolation, loneliness has also impacted the mental health of MC children and adolescents [42, 43]. Existing COVID-19 pandemic literature describes a clear link between both developmental stages' loneliness and decreased wellbeing, with a longer duration of reported loneliness being significantly associated with depression and poor mental health outcomes cross-sectionally [38, 44]. Further, increased

feelings of child loneliness from initial lockdown to the first year of the pandemic may be influenced by a decreased ability to keep daily routines, decreased social interaction, and low support for academic performance [45–47]. Findings in children and adolescents in Spring and Fall 2020 also indicate that quality parent–child relationships and support from friends were protective factors against loneliness and decreased wellbeing [48, 49]. Studies that surveyed children and adolescents individually reported risk factors for their increased loneliness and worse wellbeing during the COVID-19 pandemic, which include being female, living in a financially disadvantaged household, low connectedness to caregivers, and prior mental health difficulties [49, 50].

Thus, understanding predictors of loneliness, especially in the challenging transition from MC to adolescence, is important as access to social contexts changed during the COVID-19 pandemic, in the face of transitioning to online schooling and quarantine measures. Differences in loneliness between the two developmental stages is unclear, and understanding these differences could inform interventions to address loneliness of both MC children and adolescents, which are lacking within and beyond the COVID-19 pandemic context for both developmental stages [9, 10]. Further, although previous work has illustrated the deleterious and longitudinal effects of loneliness on adolescent mental health [51], scant work examines longitudinal, along with concurrent, contributions of multiple social contexts for MC child and adolescent loneliness within the COVID-19 pandemic.

### Present study

Past studies have illustrated that social contexts are important in both predicting and preventing MC child and adolescent loneliness [27, 52]. However, these social contexts were disrupted and restricted during the COVID-19 pandemic [53, 54]. Although early pandemic research suggests loneliness was a side effect of COVID-19-related restrictions [45, 46], little is known about how social contexts, including family environment and peers, may have longitudinally contributed to loneliness differently for MC children and adolescents during the COVID-19 pandemic. Therefore, we aimed to examine longitudinal predictors of loneliness, including COVID-19 impact, demographic characteristics, and social contexts (i.e., family environment, in-person, and virtual communication with friends) for both MC children and adolescents at two timepoints during the COVID-19 pandemic. Based on existing literature on child mental health and loneliness during and outside of the COVID-19 pandemic, we anticipated that COVID-19 pandemic changes affecting the family environment and less communication with friends would be associated with higher loneliness in MC children and adolescents. More specifically, due to increasing reliance on friendships during adolescence, we expected decreased friend communication to be a more influential predictor of adolescent loneliness than for MC children during the COVID-19 pandemic. Similarly, due to reliance of children on parents during the MC developmental stage, it was predicted family functioning would be more salient to MC child loneliness than adolescent loneliness. Lastly, to further investigate how social relationships may have mitigated or exacerbated the impact of the pandemic on loneliness in MC children and adolescents, we examined family functioning as a mediator and in-person communication and virtual communication as moderators between COVID-19 impact and loneliness in exploratory analyses.

## Materials and methods

### Procedures

The study protocol was reviewed and approved by the Institutional review Board at Nationwide Children's Hospital (approval number STUDY00001019). Study consent and assent were

implied by the completion of surveys as an approved form of consent by the Nationwide Children's Hospital Institutional Review Board. This study draws from a larger longitudinal dataset examining the impact of the COVID-19 pandemic on school-aged children and their families. Families were recruited remotely and online through pay-per-click Facebook advertisements, which were targeted to parents and were live from May 13, 2020 to July 1, 2020 (T1). These advertisements included links to online surveys on an internet-based data survey platform called Research Electronic Data Capture (REDCap). Follow-up surveys, also delivered via REDCap, were emailed 6 months later from November 30, 2020 to January 17, 2021 (T2). Participants were offered a chance to win a $100 Amazon gift card for completion of study measures at each time point. The study was approved by the Institutional Review Board, and study consent and assent were implied by completion of the T1 surveys. MC children and adolescents eligible for the study were: 8-17 years of age, fluent in English, and enrolled in school outside the home. The initial survey screened for forementioned eligibility criteria, and eligible parents reported demographic characteristics for themselves and their child via REDCap. Parents then asked their oldest willing eligible child to participate in the study and complete child measures.

## Participants

A total of 276 MC children and 177 adolescents completed T1 surveys. Accounting for attrition, 148 participants completed both T1 and T2 surveys: 92 MC children ($M_{age}$ = 10.03) and 56 adolescents ($M_{age}$ = 14.66). Descriptive statistics for MC children, adolescents, and parents are reported in Table 1. Both MC (93.5%, $n$ = 86) and adolescent (92.9%, $n$ = 52) samples were mostly White, and both samples were almost evenly divided by sex, as 57.6% ($n$ = 53) of MC children were male and 48.2% ($n$ = 27) of adolescents were male. Most parents of both developmental stages were employed, were married or had a partner, and were female. Average family income of most participating parents prior to the COVID-19 pandemic was between $100,001 and $150,000.

## Attrition analyses

Attrition analyses were conducted to compare individuals who completed both T1 and T2 measures to those who only completed T1, as there was 66.7% and 68.4% attrition in the MC child and adolescent samples, respectively. There were no significant differences between families who dropped out versus those who remained in the study on primary variables of interest, including demographic variables, COVID-19 pandemic impact, family functioning, and communication with friends (S1–S3 Tables).

## Measures

*Demographic Characteristics*: Parents completed demographic information for themselves at T1, including sex, ethnicity, race, marital/partnership status, employment, and family income. Parents also provided their participating child's demographic characteristics. Lastly, parents reported details of their child's communication with their friends (in-person and virtual friend communication).

 *COVID-19 Exposure and Family Impact Survey (CEFIS)*: At T1, children completed a modified version of the parent CEFIS, a standardized measure examining how much the COVID-19 pandemic has affected families [55]. The "impact" subscale measured the impact of the COVID-19 pandemic with 7 items on a Likert scale of 1 ("made it a lot better") to 5 ("made it a lot worse"), with higher means indicating more impact. Items more applicable to parents were removed from the original scale, such as the "ability to care for other children in your family." Items applicable to children were kept or modified, such as "how I got along with my brother

**Table 1. Sample characteristics at T1 (N = 148).**

| Baseline characteristic | Middle Childhood (*N* = 92) | Adolescence (*N* = 56) |
|---|---|---|
| | *n* (%) | *n* (%) |
| Child age in years (SD) | 10.03 (1.48) | 14.66 (1.41) |
| Sex | | |
| Male | 53 (57.6%) | 27 (48.2%) |
| Female | 39 (42.4%) | 29 (51.8%) |
| Race | | |
| White | 86 (93.5%) | 52 (92.9%) |
| Black or African American | 4 (4.3%) | 2 (3.6%) |
| Asian | 4 (4.3%) | 1 (1.8%) |
| American Indian/ Native American | 0 (0%) | 0 (0%) |
| Native Hawaiian/ Pacific Islander | 0 (0%) | 0 (0%) |
| Other | 1(1.1%) | 1 (1.8%) |
| Ethnicity | | |
| Hispanic | 8 (8.7%) | 6 (10.9%) |
| Non-Hispanic | 84 (91.3%) | 49 (89.1%) |
| Communication with friends | | |
| In-person | 17 (18.5%) | 15 (26.8%) |
| Virtual | 64 (69.6%) | 50 (89.3%) |
| Family income | | |
| Under $25,000 | 5 (5.4%) | 5 (8.9%) |
| $25,001- $50,000 | 6 (6.5%) | 8 (14.3%) |
| $50,001- $75,000 | 14 (15.2%) | 6 (10.7%) |
| $75,001- $100,000 | 18 (19.6%) | 8 (14.3) |
| $100,001- $150,000 | 25 (27.2%) | 15 (26.8%) |
| More than $150,000 | 24 (26.1%) | 14 (25.0%) |
| Parent marital/partnership status | | |
| Married or has a partner | 84 (91.3%) | 48 (85.7%) |
| Not married or no partner | 8 (8.7%) | 8 (14.3%) |

*Note.* Virtual communication (yes/no) included social media, video calling, voice/audio calling, and text messaging.

or sister." The last items on the CEFIS make the "distress" subscale, which include 2 items on a 10-point scale. The distress subscale was reduced to a 1-5 scale and was averaged with the impact scale to create an overall score of 1-5. Higher scores overall indicate higher COVID-19 pandemic impact. The measure displayed good reliability in the study sample ($\alpha$=.77).

*NIH Toolbox Item Bank v2.0 – Loneliness (Ages 8-17)*: The NIH Toolbox loneliness measure was used to assess child-reported loneliness at T1 and T2. Seven items comprise the measure and are rated on a Likert scale from never (1) to always (5). Raw scores can range from 5 to 35, with higher scores indicating more loneliness. T-scores were calculated based on a general population mean of 50 (*SD*=10). The measure has been validated and shows reliability for child self-reports [56].

*PROMIS Pediatric Item Bank v1.0 – Family Relationships – Short Form 4a*: The PROMIS Family Relationships measure was used to examine child-reported family relationships at T1. The measure includes 4 items on a Likert scale from never (1) to always (5). Raw scores can range from 4 to 25 and were converted to T-scores based on a general population mean of 50 (*SD*=10). Higher scores indicate more positive child-reported family relationships. The measure is validated and reliable for child self-report family relationship assessment [56].

## Analysis plan

Data cleaning and processing were performed prior to all analyses, including handling outliers, missing values, and testing all assumptions for regression. Descriptive and frequency statistics, correlations, attrition and sensitivity, and regression analyses were completed using IBM SPSS Version 28. Descriptive and frequency statistics were conducted on demographic characteristics. Pearson's correlations were used to examine associations between demographic variables used in the regression (control variables), communication with friends, loneliness, family functioning, and COVID-19 pandemic impact for MC children and adolescents. Attrition and sensitivity analyses were also completed using $t$-tests and chi-square tests to assess group differences between individuals who only completed T1 measures and those who completed both T1 and T2 surveys in the MC and adolescence samples. Independent samples $t$-tests and Chi-square tests examined differences between MC child and adolescent samples for variables of interest. Two separate regression models examined each developmental stage (MC children and adolescents) to analyze T1 predictors of T2 loneliness; these predictors were family income and parent marital/partnership status, which were control variables, family functioning, COVID-19 impact, in-person communication with friends, and virtual communication with friends. Lastly, to examine developmental stage's influence on social context variables that could influence loneliness, a regression model using the full sample examined the same predictor variables used in the separate MC and adolescence models, along with four developmental stage interaction variables: developmental stage x family functioning, developmental stage x COVID-19 impact, developmental stage x virtual communication with friends, and developmental stage x in-person communication with friends.

In order to investigate how social relationships may have influenced the associations between COVID-19 impact and loneliness in MC children and adolescents, mediation and moderation analyses were performed. The PROCESS macro for SPSS [57] was used to conduct a simple mediation analysis, in which COVID-19 impact was the independent variable, family functioning was the mediator, and T2 loneliness was the outcome. The same covariates that were used in the regression models were retained, which were family income and parent marital/partnership status. Because the other social relationship variables – in-person communication with friends and virtual communication with friends – were binary variables, two moderation analyses were also conducted using the PROCESS macro. COVID-19 impact was the independent variable, T2 loneliness was the dependent variable, and in-person communication and virtual communication with friends were moderators in the analyses. Again, the same covariates used in the regression model, family income and parent marital/partnership status, were retained for these models as well.

Post-hoc power analyses were conducted in G*Power to determine whether our sample size was sufficient to test our hypotheses and to help justify our results. G*Power analyses confirmed that with 92 MC children and a large effect size ($f^2 = 0.47$) [58], power was adequate (>95%) to detect multiple regressions with 6 variables. Similarly, with 56 adolescents and a large effect size, ($f^2 = 0.44$), power was adequate (>95%) to detect multiple regressions with 6 variables. Lastly, with 148 MC children and adolescents and a medium to large effect size ($f^2 = 0.34$), power was also adequate (>99%) to detect multiple regressions with 10 variables, including 4 interactions.

# Results

## Descriptive statistics and correlations

There were no significant differences found between MC children and adolescents regarding demographic variables of child race, sex, ethnicity, in-person and virtual communication with

**Table 2. Correlations between variables and T2 loneliness for MC children and adolescents.**

| Variable | 1. | 2. | 3. | 4. | 5. | 6. | 7. | 8. |
|---|---|---|---|---|---|---|---|---|
| 1. Loneliness (T2) | | -.13 | -.40** | -.38** | .35** | -.18 | -.06 | .53** |
| 2. Family income prior to COVID-19 | -.33** | | .52** | .14 | -.26 | .16 | .13 | -.14 |
| 3. Married/partnered parents (yes/no) | -.06 | .21* | | .35** | -.19 | .13 | .19 | -.22 |
| 4. Family functioning (T1) | -.33** | -.15 | .12 | | -.32* | .06 | .02 | -.40** |
| 5. COVID-19 impact (CEFIS & T1) | .37** | -.13 | -.18 | -.42** | | -.09 | -.07 | .51** |
| 6. In-person friend communication | -.25* | .18 | .05 | .11 | -.03 | | .21 | -.15 |
| 7. Virtual friend communication | -.10 | .08 | .05 | -.03 | -.09 | .13 | | .04 |
| 8. Loneliness (T1) | .59** | -.14 | -.06 | -.52** | .46** | -.21* | -.21 | |

*Note*. Correlations for adolescents are presented above the diagonal, and correlations for MC children are below the diagonal

*$p < .05$.

**$p < .01$.

friends, parent employment status, family income, parent marital/partnership status, T1 family functioning, T1 loneliness, and T1 COVID-19 pandemic impact. Mean loneliness at T1 was significantly lower for MC children ($M = 54.30$; $SD = 11.11$) than adolescents ($M = 58.64$; $SD =9.77$), $t(146) = -2.413$, $p = 0.017$, yet both had moderate levels of T1 loneliness. This difference was non-significant at T2 where MC child loneliness increased to 55.40 ($SD = 11.34$) and adolescent loneliness decreased to 56.61 ($SD = 12.31$), also indicating moderate levels for both developmental stages. The mean COVID-19 pandemic impact score at T1 was 2.93 ($SD = 0.84$) for MC children and 3.07 ($SD = 0.82$) for adolescents, both indicating moderate to high impact.

As for methods of communication with friends during the COVID-19 pandemic, we examined various modes of friend communication but only evaluated in-person and virtual communication with friends in regressions based on bivariate correlations. Adolescents reported more in-person and virtual communication than MC children. Among MC children, 18.5% maintained in-person communication with friends whereas 26.8% of adolescents maintained in-person communication with friends. Additionally, 69.6% of MC children communicated with their friends virtually as compared to 89.3% of adolescents. Both MC children ($M = 48.90$; $SD = 8.00$) and adolescents ($M = 46.94$; $SD = 7.72$) reported family functioning slightly lower than general population score norms ($M = 50$; $SD = 10$).

Point-biserial and Pearson correlation results for MC children and adolescents are in Table 2. MC child T2 loneliness was associated with higher T1 loneliness, lower family functioning, higher COVID-19 pandemic impact, not communicating with friends in-person, and lower family income prior to the COVID-19 pandemic. Adolescent T2 loneliness was associated with higher T1 loneliness, lower family functioning, higher COVID-19 pandemic impact, and having unmarried/non-partnered parents.

## Regression analyses

To examine differences in predictors of T2 loneliness by developmental stage, separate regression models were examined for MC children and adolescents. Results can be seen in Table 3. For MC children, lower family income ($b = -1.39$, $p = <.001$), lower family functioning ($b = -0.59$, $p = .01$), and higher COVID-19 impact ($b = 1.57$, $p = .04$) were significant predictors of T2 loneliness. The full MC child model was significant, $F(6,85) = 6.74$, $p = <.001$, and the MC child model explained 32.2% of variance in their loneliness. For adolescents, having non-married parents ($b = -7.03$, $p = .02$) was the only significant predictor of T2 loneliness. Higher

**Table 3. Regressions predicting full sample T2 loneliness with developmental stage as a moderator and MC child and adolescent T2 loneliness.**

| Effect | MC Children | | | | Adolescents | | | | With Interactions | | | |
|---|---|---|---|---|---|---|---|---|---|---|---|---|
| | *b* | *SE* | *β* | *p* | *b* | *SE* | *β* | *p* | *b* | *SE* | *β* | *p* |
| Intercept | 24.73 | | | <.001 | 18.53 | | | .018 | 22.72 | | | <.001 |
| Family income | -1.39 | 0.40 | -0.33 | <.001 | 0.69 | 0.60 | 0.17 | .25 | -0.66 | 0.33 | -0.16 | .057 |
| Married/partnered parents | 1.91 | 2.00 | 0.09 | .34 | -7.03 | 2.97 | -0.36 | .02 | -1.07 | 1.63 | -0.05 | .53 |
| Family functioning | -0.59 | 0.21 | -.29 | .007 | -0.43 | 0.31 | -0.19 | .17 | -0.50 | 0.18 | -0.23 | .01 |
| COVID-19 impact | 1.57 | 0.74 | 0.22 | .038 | 2.14 | 1.10 | -0.25 | .06 | 1.65 | 0.68 | 0.21 | .02 |
| In-person comm | -2.40 | 1.45 | -0.15 | .10 | -1.99 | 1.93 | -0.13 | .31 | -2.84 | 1.52 | -0.18 | .08 |
| Virtual comm | -0.67 | 1.20 | -0.05 | .58 | 0.70 | 2.78 | 0.03 | .80 | -0.71 | 1.26 | -0.05 | .59 |
| DS*family | - | - | | - | - | - | | - | -0.09 | 0.18 | -0.11 | .67 |
| DS*COVID-19 | - | - | | - | - | - | | - | 0.15 | 0.88 | 0.04 | .87 |
| DS*virtual | | | | | | | | | 1.06 | 2.72 | 0.05 | .70 |
| DS*in-person | | | | | | | | | 1.09 | 2.38 | 0.08 | .65 |

*Note*. Comm is an abbreviation for communication. DS is for developmental stage; Family in DS*Family refers to family functioning, COVID-19 in DS*COVID-19 refers to COVID-19 impact, Virtual in DS*Virtual refers to virtual communication with friends, In-person in DS*In-person refers to in-person communication with friends. Family income and married/partnered parents were control variables.

COVID-19 impact ($b$ = 2.14, $p$ = .06) was a marginally significant predictor of adolescent T2 loneliness. The full adolescent model was significant, $F(6,49)$ = 3.46, $p$ = .01, and the adolescent model also explained 29.8% of variance in their loneliness.

Using the full sample of both MC children and adolescents to examine whether MC children and adolescents differ in social context factors that influence T2 loneliness, we examined interactions of developmental stage and: friend communication, virtual communication, family functioning, and COVID-19 impact (also see Table 3). Findings displayed that in the presence of interactions, lower family income ($b$ = -0.66, $p$ = .06) and lower in-person communication with friends ($b$ = -2.84, $p$ = .08) were marginally significant predictors of higher T2 loneliness. Lower family functioning ($b$ = -0.50, $p$ = .01) and higher COVID-19 impact ($b$ = 1.65, $p$ = .02) were significant predictors for higher T2 loneliness. None of the two-way interaction terms with developmental stage were significant. The full interaction model was significant, $F(10,137)$ = 4.68, $p$ = <.001, and the model with interactions explained 25.5% of variance in loneliness.

Lastly, exploratory mediation and moderation analyses were conducted to investigate how social relationships may have influenced the impact of the COVID-19 pandemic on loneliness in both groups (S1–S3 Figs). A simple mediation revealed that family functioning was a significant mediator of COVID-19 impact on T2 loneliness for the full sample (Indirect effect 95% CI: 0.44 to 2.92) and MC children (Indirect effect 95% CI: 0.09 to 3.04). However, it was a not a significant mediator for adolescents (Indirect effect 95% CI: -0.08 to 4.36). Moderation analyses, which tested in-person and virtual communication as moderators between COVID-19 impact on T2 loneliness, revealed no significant findings.

## Discussion

When comparing predictors of loneliness overtime during MC and adolescence, our study found family functioning, COVID-19 impact, and family income significantly influenced MC child loneliness. On the other hand, in our sample, only parent marital/partnership status significantly influenced adolescent loneliness over time, along with COVID-19 impact being a marginally significant influence on adolescent loneliness. As no social context factors (family

functioning nor friend communication) significantly influenced adolescent loneliness, the direct impact of the COVID-19 pandemic and parent partnership status may outweigh the influence of most other factors and social contexts. Exploratory mediation analyses also revealed processes by which COVID-19 impact may influence family functioning, which may subsequently influence loneliness longitudinally in both groups, but more specifically in MC children.

Our study found that worse family functioning was a significant predictor of higher loneliness in MC children, even while taking into consideration their reports of COVID-19 impact. This finding was seen in both regression and exploratory mediation analyses. Although MC children may increasingly interact with others outside the home, healthy family functioning and quality parent–child relationships are important to mitigate feelings of loneliness and other poor mental health outcomes [14, 15]. However, existing healthy family functioning may have been negatively affected due to the COVID-19 pandemic [59]. Attributable to COVID-19 pandemic related changes, such as financial strain and household reorganization, children 10-18 years old have reported increased family conflict at home and higher perceived parent stress [40]. Consequently, poor parent–child relationships during the COVID-19 pandemic have been associated with loneliness reported by children 11-20 years old [60]. This finding reiterates the importance of healthy family functioning, which may be a key point of intervention to mitigate MC child loneliness during times of stress and rapid change.

Lower family income was another significant predictor of MC child loneliness in our study. Varied results exist pertaining to the role of income in MC loneliness during the COVID-19 pandemic, with some indicating no relationship [61] and others indicating low income as linked to worse wellbeing [62]. Along with the family context and the context of the COVID-19 pandemic, MC child loneliness may be more susceptible to income-related forms of hardship. The influence of low-income on MC child loneliness may be attributable to parent employment and income instability leading to decreased parent mental health, which may connect to an increased risk of child stress and decreased child wellbeing [61, 62]. Focusing on ameliorating stress of both children and parents in at-risk families may be beneficial in reducing loneliness through providing families with temporary financial assistance and promoting child wellbeing through outside of school and after school activities [63].

Among both the MC and adolescent groups, higher self-reported COVID-19 impact was a significant and marginally significant predictor of increased loneliness, respectively. This aligns with previous literature, which describes the negative impacts of the COVID-19 pandemic on MC and adolescent loneliness and wellbeing [45–47]. These study results add to the growing body of literature [36, 51] that demonstrate the damaging impacts of the COVID-19 pandemic on child wellbeing over time and calls for more widely available child mental health screening post-pandemic.

As adolescents begin to increasingly socialize in contexts outside the home, such as school and extracurricular activities, friendship also becomes necessary for their wellbeing [12]. Thus, due to adolescents relying more on friends for support than MC children [25], it was hypothesized negative changes to friend communication would be more salient for adolescent loneliness during the COVID-19 pandemic. However, this hypothesis was not reflected in the adolescent model, in which no social context factors emerged as significant. This may indicate that in the presence of COVID-19, social contexts may be less important than the direct impact of the COVID-19 pandemic for adolescent loneliness. Alternatively, our smaller sample of adolescents relative to the overall sample may have prevented us from detecting effects of social contexts on adolescent loneliness. Along with in-person communication, virtual communication also did not significantly influence adolescent loneliness, and some literature has pointed to this null finding of social technology not being related to adolescent wellbeing changes

during the COVID-19 pandemic [64]. This may also point to the mixed findings of media use on adolescent wellbeing during the COVID-19 pandemic [65, 66].

A significant predictor of adolescent T2 loneliness was non-partnered parents. Past studies, both within [67] and outside of the COVID-19 pandemic [68], support adolescents with non-partnered or separated parents reporting higher loneliness. This is also consistent with literature on parental divorce predicting poor emotional outcomes [69], as well as family structure changes predicting weaker parent–child emotional ties, which are linked to increased loneliness [70]. Therefore, in addition to the context of the COVID-19 pandemic, adolescent loneliness may be more susceptible to single-parent households. This finding may call for special attention and specialized intervention work that serves adolescents with non-traditional family structures in times of crisis.

Although different social contexts and demographic characteristics significantly predicted increased loneliness for MC children and adolescents, there were no significant interaction effects between developmental stage and social context factors on loneliness. Therefore, our results did not support statistical differences in family functioning, friend communication, or COVID-19 impact by developmental stage. This may be due to sample size, which might have been insufficient to detect interactions in regression models and highlights the need for future research with a larger sample to examine developmental stage differences in influences of social contexts on loneliness.

## Limitations

The results of this study must be viewed considering its limitations. Like most longitudinal studies, attrition was expected; as the study was set in summer and winter of 2020, when most children attended school online. This could have led to increased participant dropout due to life changes and increased hardship. However, attrition analyses found no significant differences between participants who completed both T1 and T2 surveys, compared to those who dropped out. Additionally, it should also be noted the sample was largely White and non-Hispanic, and more than half of the sample reported a family income over $100,000. Therefore, enrolled families may be more socioeconomically privileged than most Americans, and findings may not generalize to the wider American public or international populations. Additionally, we did not screen for preexisting psychological disorders, so it is unclear the degree to which preexisting child mental health statuses may have influenced our results. Due to the constraints of the pandemic, recruitment was conducted through Facebook, which comes with disadvantages, such as sampling and response bias [71]. Future studies that use social media recruitment strategies should better prioritize representation in their sample; however, it should be noted that zip codes of various socioeconomic backgrounds were targeted in this study to attempt to improve diversity [72]. Lastly, it is likely none of our interactions between developmental stage and social context variables emerged as significant due to our inadequate sample size [73], and we were only able to explore potential mediation processes as our study only spanned two timepoints.

## Future directions

For MC children, supporting positive family functioning may help protect against loneliness and negative mental health outcomes within and outside the context of the COVID-19 pandemic. This may be done through family-based interventions that strengthen relationships and improve communication among family members. Teletherapy interventions that involve parent training, teach parents how to cope with anxiety, and help children manage their stress have been successful in strengthening family functioning and relationships, along with

improving child and parent mental health outcomes during the pandemic [74, 75]. Outside of the COVID-19 pandemic, positive psychology and family interaction interventions have also been effective in increasing family happiness, empowering families with positive practices, and strengthening family relationships [76, 77]. Developing interventions targeted towards preventing poor MC child mental health outcomes in the face of sudden changes and adversity, such as the COVID-19 pandemic, future pandemics, or possible lockdowns, especially in families of lower socioeconomic status, may also be helpful. This may be true to also prevent poor adolescent mental health outcomes in face of sudden changes, especially in families with non-traditional family structures.

## Conclusion

Limited literature exists on understanding how social context factors, such as family and friendships, can influence loneliness in MC children versus adolescents during the COVID-19 pandemic. Further, almost no studies examine these forementioned associations longitudinally, making it difficult to understand how loneliness could be mitigated over time. Therefore, this study examined and compared longitudinal predictors of loneliness, including social contexts and COVID-19 pandemic impact, in both MC and adolescence. Results show that worse family functioning remained a significant predictor of higher loneliness only in MC children. Contrary to our hypothesis that changes in friend communication would be more salient to influencing loneliness in adolescents, the only significant predictor of increased adolescent loneliness was having non-partnered parents, along with self-reported higher COVID-19 impact being a marginally significant predictor. The results support future interventions focusing on optimizing family functioning to alleviate loneliness among middle childhood and adolescent populations during the pandemic.

## Supporting information

**S1 Fig. Simple mediation model of T1 family functioning as a mediator between T1 COVID-19 impact and T2 loneliness.**
(TIF)

**S2 Fig. T1 Family Functioning as a Mediator Between T1 COVID-19 Impact and T2 Loneliness in the Full Sample.**
(TIF)

**S3 Fig. T1 family functioning as a mediator between T1 COVID-19 impact and T2 loneliness in MC children.**
(TIF)

**S1 Table. Comparisons on variables of interest for children and adolescents present at only T1 versus both T1 and T2.**
(DOCX)

**S2 Table. Examining differences in children present at just T1 and both T1 and T2 through chi-square tests.**
(DOCX)

**S3 Table. Examining differences in adolescents present at just T1 and both T1 and T2 through chi-square tests.**
(DOCX)

## Acknowledgments

We would like to thank all the families who agreed to participate in the study and supported the study by providing their data.

## Author Contributions

**Conceptualization:** Ashley Benhayoun, Anna Olsavsky, Micah A. Skeens.

**Data curation:** Micah A. Skeens.

**Formal analysis:** Ashley Benhayoun, Anna Olsavsky.

**Funding acquisition:** Micah A. Skeens.

**Investigation:** Micah A. Skeens.

**Methodology:** Micah A. Skeens.

**Project administration:** Micah A. Skeens.

**Resources:** Micah A. Skeens.

**Software:** Anna Olsavsky, Micah A. Skeens.

**Supervision:** Anna Olsavsky, Terrah Foster Akard, Cynthia Gerhardt, Micah A. Skeens.

**Validation:** Micah A. Skeens.

**Visualization:** Ashley Benhayoun, Anna Olsavsky, Micah A. Skeens.

**Writing – original draft:** Ashley Benhayoun, Anna Olsavsky.

**Writing – review & editing:** Ashley Benhayoun, Anna Olsavsky, Terrah Foster Akard, Cynthia Gerhardt, Micah A. Skeens.

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
