## [Decision Letter · Decision Letter 0]

14 Jun 2024

PONE-D-24-14891

Predictors of Loneliness among Middle Childhood and Adolescence during the COVID-19 Pandemic

PLOS ONE

Dear Dr. Skeens,

Thank you for submitting your manuscript to PLOS ONE. I have now received evaluations of your manuscript from two expert reviewers, both of whom are well-qualified to review the work. After careful consideration, we believe the manuscript has merit but does not fully meet PLOS ONE’s publication criteria in its current form. Therefore, we invite you to submit a revised version of the manuscript that addresses the points raised during the review process.

As you will see from the attached comments, both reviewers recognize the significant potential of your manuscript. However, they have also identified some issues that need to be addressed. I am hopeful that you will be able to revise the work to address these concerns.

Both reviewers are very clear about their points, so I won’t repeat them here. Many of the comments are relatively minor and could easily be addressed by carefully revising the paper to increase clarity. Other issues might require more careful consideration.

I will try to summarize the main points you need to focus on:

Both reviewers highlight the issue of the small sample size. I do not agree with increasing the sample size, given the nature of the data and the historical period during which they were collected. Expanding the sample now and retesting the same analyses would undermine the results. Instead, I suggest conducting a post-hoc power analysis to demonstrate whether your sample is sufficient (or is not) to test your hypotheses and justify your results.Reviewer #1 recommends conducting additional analyses, such as testing the interaction between the impact of COVID-19 and some predictors of loneliness, like family functioning. However, the correlation between these two variables (-.42**) would violate one of the assumptions of moderation models, which is that the predictor and moderator should not (preferably) be correlated. Nonetheless, I agree with the reviewer's suggestion to conduct additional analyses to provide a more comprehensive picture of your study. Therefore, I suggest evaluating possible longitudinal mediation effects (e.g., predictor and mediator at T1 and outcome at T2).Last but not least, I strongly encourage you to upload your data to a data repository, such as the Open Science Framework (OSF). This is a necessary condition for your manuscript to be accepted for publication. See the PLOS ONE submission guidelines for more information (https://journals.plos.org/plosone/s/submission-guidelines#loc-guidelines-for-specific-study-types).

We look forward to receiving your revised manuscript.

Kind regards,

Silvana Mula, Ph.D.

Academic Editor

PLOS ONE

2. In this instance it seems there may be acceptable restrictions in place that prevent the public sharing of your minimal data. However, in line with our goal of ensuring long-term data availability to all interested researchers, PLOS’ Data Policy states that authors cannot be the sole named individuals responsible for ensuring data access (http://journals.plos.org/plosone/s/data-availability#loc-acceptable-data-sharing-methods).

Reviewers' comments:

Reviewer's Responses to Questions

**Comments to the Author**

1. Is the manuscript technically sound, and do the data support the conclusions?

Reviewer #1: Partly

Reviewer #2: Yes

2. Has the statistical analysis been performed appropriately and rigorously?

Reviewer #1: No

Reviewer #2: Yes

3. Have the authors made all data underlying the findings in their manuscript fully available?

Reviewer #1: No

Reviewer #2: Yes

4. Is the manuscript presented in an intelligible fashion and written in standard English?

Reviewer #1: Yes

Reviewer #2: Yes

5. Review Comments to the Author

Reviewer #1: Thank you for the opportunity to review this paper that examined the predictors of middle childhood and adolescent loneliness during the Covid-19 pandemic. However, I have some major concerns with the current paper. These are described below.

*My biggest issue with the paper concerns the sample size. I am concerned that the authors did not have an adequate sample size to estimate the hypothesized model, even including children and adolescents. Therefore, it is crucial that the authors test the robustness of the results with a sensitivity analysis.

*In the 'Regression Analyses' section, it appears that the authors reported unstandardized coefficients instead of β coefficients.

* Why did the authors not test the interaction effect between the impact of COVID-19 and social predictors of loneliness (e.g., family functioning)

* Using the impact of the COVID-19 pandemic as a covariate in the model does not seem to provide novel evidence about the predictors of loneliness during childhood and adolescence. While controlling for this variable may confirm the relationship between the predictors and the independent variable during the health crisis, it does not provide information on how the pandemic affected social contexts that influence feelings of loneliness.

Therefore, according to a previous cross-sectional study (Skeens et al., 2023), I suggest to investigate how social relationships may have mitigated or exacerbated the impact of the pandemic on loneliness in middle children and adolescents.

Reviewer #2: This paper investigates the predictors of loneliness among middle childhood (8-12 years) and adolescent (13-17 years) populations during the COVID-19 pandemic through a longitudinal study, with a focus on social background and pandemic impact. The paper employs a rigorous research design and appropriate data analysis methods, yielding conclusions with practical significance. Overall, this paper features a novel topic, rigorous methodology, and results that hold significant theoretical and practical implications. While there are areas that could be further optimized, it remains a high-quality research paper.

Here are some specific suggestions for improvement:

1. Abstract: The abstract lacks a detailed description of the research methods, such as the types of questionnaires used.

2. Sample Size: The study includes a relatively small sample size. It is recommended to expand the sample size.

3. Sample Selection: Apply stricter criteria for sample selection, such as excluding participants with psychological disorders.

4. Data Collection Process: Provide more details about the data collection process. Specify how the questionnaires were distributed (online or offline) and the timeframe for participants to complete them.

5. Data Cleaning and Processing: Explain whether data cleaning and processing were performed prior to analysis, including handling outliers and missing values.

6. Control Variables: Indicate whether control variables were included in the linear regression model, and if so, list the variables controlled for.

7. Practical Implications: In the conclusion section, highlight the practical applications of the findings. For instance, "The results support future interventions focusing on optimizing family functioning to alleviate loneliness among middle childhood and adolescent populations during the pandemic."

6. PLOS authors have the option to publish the peer review history of their article (what does this mean?). If published, this will include your full peer review and any attached files.

**Do you want your identity to be public for this peer review?** For information about this choice, including consent withdrawal, please see our Privacy Policy.

Reviewer #1: No

Reviewer #2: **Yes: **Wenfei Zhu

---

## [Author Response · Author response to Decision Letter 0]

26 Jun 2024

Thank you for the opportunity to revise and resubmit our manuscript, “Predictors of Loneliness among Middle Childhood and Adolescence during the COVID-19 Pandemic.” We found the comments very useful and that they strengthened the paper, especially methodologically! Please see our point-by-point response to reviewers’ feedback below. 

As you will see from the attached comments, both reviewers recognize the significant potential of your manuscript. However, they have also identified some issues that need to be addressed. I am hopeful that you will be able to revise the work to address these concerns. Both reviewers are very clear about their points, so I won’t repeat them here. Many of the comments are relatively minor and could easily be addressed by carefully revising the paper to increase clarity. Other issues might require more careful consideration.

We appreciate the kind words regarding the potential of our manuscript. We are happy to respond to reviewer feedback. 

I will try to summarize the main points you need to focus on:

1. Both reviewers highlight the issue of the small sample size. I do not agree with increasing the sample size, given the nature of the data and the historical period during which they were collected. Expanding the sample now and retesting the same analyses would undermine the results. Instead, I suggest conducting a post-hoc power analysis to demonstrate whether your sample is sufficient (or is not) to test your hypotheses and justify your results.

Thank you for this suggestion. Separate power analyses were conducted in G*Power to determine our achieved power given our sample size and identified effect sizes across focal regression models in MC children, adolescents, and the full sample, including interaction terms. The analyses revealed power was adequate, above 95%, for all regressions. Further explanation has been added to the end of the methods section.

2. Reviewer #1 recommends conducting additional analyses, such as testing the interaction between the impact of COVID-19 and some predictors of loneliness, like family functioning. However, the correlation between these two variables (-.42**) would violate one of the assumptions of moderation models, which is that the predictor and moderator should not (preferably) be correlated. Nonetheless, I agree with the reviewer's suggestion to conduct additional analyses to provide a more comprehensive picture of your study. Therefore, I suggest evaluating possible longitudinal mediation effects (e.g., predictor and mediator at T1 and outcome at T2).

Mediations and moderations are reported in methods and results. We conducted mediation and moderation analyses using Hayes (2017) SPSS PROCESS Macro (version 4.3.1). The PROCESS macro does not allow for binary mediators, so we opted to conduct moderation instead of mediation for in-person and virtual communication. Only the mediation models that explored T1 family functioning as a mediator between T1 COVID-19 Impact and T2 Loneliness were significant in the full sample and MC children. This same model for adolescents, as well as all moderation analyses examining both communication methods as moderators between T1 COVID-19 Impact and T2 Loneliness for the full sample, MC children, and adolescents were not significant. We have attached mediation figures in “S1, S2, S3 figures.pptx”

3. Last but not least, I strongly encourage you to upload your data to a data repository, such as the Open Science Framework (OSF). This is a necessary condition for your manuscript to be accepted for publication. See the PLOS ONE submission guidelines for more information (https://journals.plos.org/plosone/s/submission-guidelines#loc-guidelines-for-specific-study-types). 

Thank you for this recommendation. We uploaded our data to OSF, which includes a datafile and data dictionary. The link is here: https://osf.io/8fec4/

Reviewer #1: Thank you for the opportunity to review this paper that examined the predictors of middle childhood and adolescent loneliness during the Covid-19 pandemic. However, I have some major concerns with the current paper. These are described below.

*My biggest issue with the paper concerns the sample size. I am concerned that the authors did not have an adequate sample size to estimate the hypothesized model, even including children and adolescents. Therefore, it is crucial that the authors test the robustness of the results with a sensitivity analysis. 

Thank you for the opportunity to further investigate our sample size. Based on the recommendation of the editor, we conducted post hoc power analyses in G*Power, which are now reported in the methods section. The analyses found greater than 95% power across all analyses. Sensitivity analyses are attached to the submission as “S1 Table, S2 Table, and S3 Table.”

*In the 'Regression Analyses' section, it appears that the authors reported unstandardized coefficients instead of β coefficients. 

Both have now been reported in the updated Table 3, and the notation has been updated to reflect unstandardized coefficients in-text in the ‘Regression Analyses’ section.

* Why did the authors not test the interaction effect between the impact of COVID-19 and social predictors of loneliness (e.g., family functioning) 

Thank you for the suggestion. We performed and now have included mediation analyses of family functioning as a mediator between COVID-19 impact and T2 loneliness. This is further explained at the end of the results section and can be seen in the new file, “S1, S2, S3 figures.”

* Using the impact of the COVID-19 pandemic as a covariate in the model does not seem to provide novel evidence about the predictors of loneliness during childhood and adolescence. While controlling for this variable may confirm the relationship between the predictors and the independent variable during the health crisis, it does not provide information on how the pandemic affected social contexts that influence feelings of loneliness.

Therefore, according to a previous cross-sectional study (Skeens et al., 2023), I suggest to investigate how social relationships may have mitigated or exacerbated the impact of the pandemic on loneliness in middle children and adolescents. 

To explore how social relationships may have mitigated or exacerbated the impact of the pandemic on loneliness, we have performed and reported mediation and moderation analyses in the methods and results sections. More specifically, we explored family functioning as a mediator and both in-person communication and virtual communication as moderators in the full sample, and MC children and adolescents separately as well.

Reviewer #2: This paper investigates the predictors of loneliness among middle childhood (8-12 years) and adolescent (13-17 years) populations during the COVID-19 pandemic through a longitudinal study, with a focus on social background and pandemic impact. The paper employs a rigorous research design and appropriate data analysis methods, yielding conclusions with practical significance. Overall, this paper features a novel topic, rigorous methodology, and results that hold significant theoretical and practical implications. While there are areas that could be further optimized, it remains a high-quality research paper.

Thank you for your positive feedback on our manuscript.

Here are some specific suggestions for improvement:

1. Abstract: The abstract lacks a detailed description of the research methods, such as the types of questionnaires used. 

Information on questionnaires used has now been added to the abstract

2. Sample Size: The study includes a relatively small sample size. It is recommended to expand the sample size. 

Unfortunately, we are unable to collect more data given that the period of time data was collected is so central to the study’s interpretation. Therefore, based on the recommendation of the editor, we have performed and reported on power analyses in the methods section. 

3. Sample Selection: Apply stricter criteria for sample selection, such as excluding participants with psychological disorders. 

Thank you for raising this issue. Our screening criteria for this study included: MC children and adolescents being 8-17 years of age, fluent in English, and enrolled in school outside the home. In future studies we will make sure to include stricter criteria, such as excluding individuals with psychological disorders and neurodevelopmental delays as you mentioned. We have addressed this in our limitations section.

4. Data Collection Process: Provide more details about the data collection process. Specify how the questionnaires were distributed (online or offline) and the timeframe for participants to complete them. 

Thank you for the opportunity to clarify. We agree our language could have been clearer. We have updated the text in the procedures section in the methods section to include details on questionnaires being distributed online and the time given to participants to complete surveys.

5. Data Cleaning and Processing: Explain whether data cleaning and processing were performed prior to analysis, including handling outliers and missing values. 

An explanation has been added to the beginning of the analysis plan section in methods: “Data cleaning and processing were performed prior to all analyses, including handling outliers, missing values, and testing all assumptions for regression.”

6. Control Variables: Indicate whether control variables were included in the linear regression model, and if so, list the variables controlled for. 

A clarification of control variables used has been added to the methods section, as well as a note under table 3. 

7. Practical Implications: In the conclusion section, highlight the practical applications of the findings. For instance, “The results support future interventions focusing on optimizing family functioning to alleviate loneliness among middle childhood and adolescent populations during the pandemic.” 

Thank you for your comment. We agree with your practical assessment of our findings and have added your sentence to the conclusion section.

---

## [Decision Letter · Decision Letter 1]

17 Jul 2024

Predictors of Loneliness among Middle Childhood and Adolescence during the COVID-19 Pandemic

PONE-D-24-14891R1

Dear Dr. Skeens,

We are pleased to inform you that your manuscript has been judged scientifically suitable for publication and will be formally accepted for publication once it meets all outstanding technical requirements. As you can see, both reviewers are satisfied with the integrations made and with your responses to their comments. 

Within one week, you will receive an e-mail detailing the required amendments. When these have been addressed, you’ll receive a formal acceptance letter and your manuscript will be scheduled for publication.

Kind regards,

Silvana Mula, Ph.D.

Academic Editor

PLOS ONE

Additional Editor Comments (optional):

Reviewers' comments:

Reviewer's Responses to Questions

**Comments to the Author**

1. If the authors have adequately addressed your comments raised in a previous round of review and you feel that this manuscript is now acceptable for publication, you may indicate that here to bypass the “Comments to the Author” section, enter your conflict of interest statement in the “Confidential to Editor” section, and submit your "Accept" recommendation.

Reviewer #1: All comments have been addressed

Reviewer #2: All comments have been addressed

2. Is the manuscript technically sound, and do the data support the conclusions?

Reviewer #1: Yes

Reviewer #2: Yes

3. Has the statistical analysis been performed appropriately and rigorously? 

Reviewer #1: Yes

Reviewer #2: Yes

4. Have the authors made all data underlying the findings in their manuscript fully available?

Reviewer #1: Yes

Reviewer #2: Yes

5. Is the manuscript presented in an intelligible fashion and written in standard English?

Reviewer #1: Yes

Reviewer #2: Yes

6. Review Comments to the Author

Reviewer #1: (No Response)

Reviewer #2: (No Response)

7. PLOS authors have the option to publish the peer review history of their article (what does this mean?). If published, this will include your full peer review and any attached files.

Reviewer #1: No

Reviewer #2: **Yes: **Wenfei Zhu

---

## [Editor Report · Acceptance letter]

6 Aug 2024

PONE-D-24-14891R1 

PLOS ONE

Dear Dr. Skeens, 

I'm pleased to inform you that your manuscript has been deemed suitable for publication in PLOS ONE. Congratulations! Your manuscript is now being handed over to our production team.

Kind regards, 

on behalf of

Dr. Silvana Mula 

Academic Editor

PLOS ONE